# EFFICIENT ACTIVE SEARCH FOR COMBINATORIAL OPTIMIZATION PROBLEMS

**André Hottung**
Bielefeld University, Germany
`andre.hottung@uni-bielefeld.de`

**Yeong-Dae Kwon**
Samsung SDS, Korea
`y.d.kwon@samsung.com`

**Kevin Tierney**
Bielefeld University, Germany
`kevin.tierney@uni-bielefeld.de`

## ABSTRACT

Recently, numerous machine learning based methods for combinatorial optimization problems have been proposed that learn to construct solutions in a sequential decision process via reinforcement learning. While these methods can be easily combined with search strategies like sampling and beam search, it is not straightforward to integrate them into a high-level search procedure offering strong search guidance. Bello et al. (2016) propose active search, which adjusts the weights of a (trained) model with respect to a single instance at test time using reinforcement learning. While active search is simple to implement, it is not competitive with state-of-the-art methods because adjusting all model weights for each test instance is very time and memory intensive. Instead of updating all model weights, we propose and evaluate three efficient active search strategies that only update a subset of parameters during the search. The proposed methods offer a simple way to significantly improve the search performance of a given model and outperform state-of-the-art machine learning based methods on combinatorial problems, even surpassing the well-known heuristic solver LKH3 on the capacitated vehicle routing problem. Finally, we show that (efficient) active search enables learned models to effectively solve instances that are much larger than those seen during training.

## 1 INTRODUCTION

In recent years, a wide variety of machine learning (ML) based methods for combinatorial optimization problems have been proposed (e.g., Kool et al. (2019); Hottung et al. (2020)) . While early approaches failed to outperform traditional operations research methods, the gap between handcrafted and learned heuristics has been steadily closing. However, the main potential of ML-based methods lies not only in their ability to outperform existing methods, but in automating the design of customized heuristics in situations where no handcrafted heuristics have yet been developed. We hence focus on developing approaches that require as little additional problem-specific knowledge as possible.

Existing ML based methods for combinatorial optimization problems can be classified into construction methods and improvement methods. Improvement methods search the space of complete solutions by iteratively refining a given start solution. They allow for a guided exploration of the search space and are able to find high-quality solutions. However, they usually rely on problem-specific components. In contrast, construction methods create a solution sequentially starting from an empty solution (i.e., they consider a search space consisting of incomplete solutions). At test time, they can be used to either greedily construct a single solution or to sample multiple solutions from the probability distribution encoded in the trained neural network. Furthermore, the sequential solution generation process can be easily integrated into a beam search without requiring any problem-specific components. However, search methods like sampling and beam search offer no (or very limited) search guidance. Additionally, these methods do not react towards the solutions seen so far, i.e., the underlying distribution from which solutions are sampled is never changed throughout the search.

Bello et al. (2016) propose a generic search strategy called active search that allows an extensive, guided search for construction methods without requiring any problem specific components. Active search is an iterative search method that at each iteration samples solutions for a single test instance using a given model and then adjusts the parameters of that model with the objective to increase the likelihood of generating high-quality solutions in future iterations. They report improved performance over random sampling when starting the search from an already trained model. Despite promising results, active search has not seen adaption in the literature. The reason for this is its resource requirements, as adjusting all model parameters separately for each test instance is very time intensive, especially compared to methods that can sample solutions to multiple different instances in one batch.

We extend the idea of active search as follows. (1) We propose to only adjust a subset of (model) parameters to a single instance during the search, while keeping all other parameters fixed. We show that this efficient active search (EAS) drastically reduces the runtime of active search without impairing the solution quality. (2) We implement and evaluate three different implementations of EAS and show that all offer significantly improved performance over pure sampling approaches.

In our EAS implementations, the majority of (model) parameters are not updated during the search, which drastically reduces the runtime, because gradients only need to be computed for a subset of model weights, and most operations can be applied identically across a batch of different instances. Furthermore, we show that for some problems, EAS finds even better solutions than the original active search. All EAS implementations can be easily applied to existing ML construction methods.

We evaluate the proposed EAS approaches on the traveling salesperson problem (TSP), the capacitated vehicle routing problem (CVRP) and the job shop scheduling problem (JSSP). For all problems, we build upon already existing construction approaches that only offer limited search capabilities. In all experiments, EAS leads to significantly improved performance over sampling approaches. For the CVRP and the JSSP, the EAS approaches outperform all state-of-the-art ML based approaches, and even the well-known heuristic solver LKH3 for the CVRP. Furthermore, EAS approaches assists in model generalization, resulting in drastically improved performance when searching for solutions to instances that are much larger than the instances seen during model training.

## 2   Literature review

**Construction methods**    Hopfield (1982) first used a neural network (a Hopfield network) to solve small TSP instances with up to 30 cities. The development of recent neural network architectures has paved the way for ML approaches that are able to solve large instances. The pointer network architecture proposed by Vinyals et al. (2015) efficiently learns the conditional probability of a permutation of a given input sequence, e.g., a permutation of cities for a TSP solution. The authors solve TSP instances with up to 50 cities via supervised learning. Bello et al. (2016) report that training a pointer network via actor-critic RL instead results in a better performance on TSP instances with 50 and 100 cities. Furthermore, graph neural networks are used to solve the TSP, e.g., a graph embedding network in Khalil et al. (2017) and a graph attention network in Deudon et al. (2018).

The first applications of neural network based methods to the CVRP are reported by Nazari et al. (2018) and Kool et al. (2019). Nazari et al. (2018) propose a model with an attention mechanism and a recurrent neural network (RNN) decoder that can be trained via actor-critic RL. Kool et al. (2019) propose an attention model that uses an encoder that is similar to the encoder used in the transformer architecture Vaswani et al. (2017). Peng et al. (2019) and Xin et al. (2021) extend the attention model to update the node embeddings throughout the search, resulting in improved performance at the cost of longer runtimes for the CVRP. Falkner & Schmidt-Thieme (2020) propose an attention-based model that constructs tours in parallel for the CVRP with time windows.

While ML-based construction methods have mainly focused on routing problems, there are some notable exceptions. For example, Khalil et al. (2017) use a graph embedding network approach to solve the minimum vertex cover and the maximum cut problems (in addition to the TSP). Zhang et al. (2020) propose a graph neural network based approach for the job shop scheduling problem (JSSP). Li et al. (2018) use a guided tree search enhanced ML approach to solve the maximal independent set, minimum vertex cover, and the maximal clique problems. For a more detailed review of ML methods on different combinatorial optimization problems, we refer to Vesselinova et al. (2020).

While most approaches construct routing problem solutions autoregressively, some approaches predict a heat-map that describes which edges will likely be part of a good solution. The heat-map is then used in a post-hoc search to construct solutions. Joshi et al. (2019) use a graph convolutional network to create a heat-map and a beam search to search for solutions. Similarly, Fu et al. (2020) use a graph convolutional residual network with Monte Carlo tree search to solve large TSP instances. Kool et al. (2021) use the model from Joshi et al. (2019) to generate the heat-map and use it to search for solution to TSP and CVRP instances with a dynamic programming based approach.

**Improvement methods** Improvement methods integrate ML based methods into high-level search heuristics or try to learn improvement operators directly. In general, they often invest more time into solving an instance than construction based methods (and usually find better solutions). Chen & Tian (2019) propose an approach that iteratively changes a local part of the solution. At each iteration, the trainable region picking policy selects a part of the solution that should be changed and a trainable rule picking policy selects an action from a given set of possible modification operations. Hottung & Tierney (2020) propose a method for the CVRP that iteratively destroys parts of a solution using predefined, handcrafted operators and then reconstructs them with a learned repair operator. Wu et al. (2021) and de O. da Costa et al. (2020) propose to use RL to pick an improving solution from a specified local neighborhood (e.g., the 2-Opt neighborhood) to solve routing problems. Hottung et al. (2021) learn a continuous representation of discrete routing problem solutions using conditional variational autoencoders and search for solutions using a generic, continuous optimizer.

## 3 Solving combinatorial optimization problems with EAS

We propose three EAS implementations that adjust a small subset of (model) parameters in an iterative search process. Given an already trained model, we investigate adjusting (1) the normally static embeddings of the problem instance that are generated by the encoder model, (2) the weights of additional instance-specific residual layers added to the decoder, and (3) the parameters of a lookup table that directly affect the probability distribution returned by model. In each iteration, multiple solutions are sampled for one instance and the dynamic (model) parameters are adjusted with the goal of increasing the probability of generating high quality solutions (as during model training). This allows the search to sample solutions of higher quality in subsequent iterations, i.e., the search can focus on the more promising areas of the search space. Once a high-quality solution for an instance is found, the adjusted parameters are discarded, so that the search process can be repeated on other instances. All strategies efficiently generate solutions to a batch of instances in parallel, because the network layers not updated during the search are applied identically to all instances of the batch.

**Background** RL based approaches for combinatorial problems aim to learn a neural network based model $p_\theta(\pi|l)$ with weights $\theta$ that can be used to generate a solution $\pi$ given an instance $l$. State-of-the-art approaches usually use a model that consists of an encoder and a decoder unit. The encoder usually creates static embeddings $\omega$ that describe the instance $l$ using a computationally expensive encoding process (e.g., Kool et al. (2019); Kwon et al. (2020)). The static embeddings are then used to autoregressively construct solutions using the decoder over $T$ time steps. At each step $t$, the decoder $q_\phi(a|s_t, \omega)$, with weights $\phi \subset \theta$, outputs a probability value for each possible action $a$ in the state $s_t$ (e.g., for the TSP, each action corresponds to visiting a different city next). The starting state $s_1$ describes the problem instance $l$ (e.g., the positions of the cities for the TSP and the starting city) and the state $s_{t+1}$ is obtained by applying the action $a_t$ selected at time step $t$ to the state $s_t$. The (partial) solution $\pi_t$ is defined by the sequence of selected actions $a_1, a_2, \ldots, a_t$. Once a complete solution, $\pi_T$, fulfilling all constraints of the problem is constructed, the objective function value $C(\pi, l)$ of the solution can be computed (e.g., the tour length for the TSP).

Figure 1 shows the solution generation for a TSP instance with a model that uses static embeddings. The static embeddings $\omega$ are used at each decoding step to generate a probability distribution over all possible next actions and the selected action is provided to the decoder in the next decoding step. During testing, solutions can be constructed by either selecting actions greedily or by sampling each action according to $q_\phi(a|s_t, \omega)$. Since the static embeddings are not updated during solution generation they only need to be computed once per instance, which allows to quickly sample multiple solutions per instance. We note that not all models use static embeddings. Some approaches update all instance embeddings after each action (e.g., Zhang et al. (2020)), which allows the embeddings to contain information on the current solution state $s_t$.

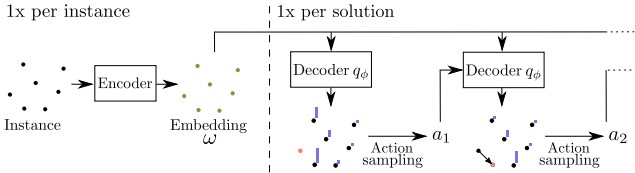

Figure 1: Sampling a solution for the TSP with a model $p_\theta(\pi|l)$ that uses static instance embeddings.

### 3.1 EMBEDDING UPDATES

Our first proposed strategy, called EAS-Emb, updates the embeddings $\omega$ generated by an encoder using a loss function consisting of an RL component $\mathcal{L}_{RL}$ and an imitation learning (IL) component $\mathcal{L}_{IL}$. The loss $\mathcal{L}_{RL}$ is based on REINFORCE (Williams, 1992) and is the expected cost of the generated solutions, $\mathbb{E}[C(\pi)]$. We aim to adjust the embedding parameters to increase the likelihood of generating solutions with lower costs (e.g., a shorter tour length for the TSP). The loss $\mathcal{L}_{IL}$ is the negation of the log-probability of (re-)generating the best solution seen so far. We adjust the embedding parameters to increase this probability.

More formally, for an instance $l$ we generate the embeddings $\omega$ using a given encoder. Based on $\omega$, we can (repeatedly) sample a solution $\pi$ whose cost is $C(\pi)$. A subset of the embeddings $\hat{\omega} \subseteq \omega$ is adjusted to minimize $\mathcal{L}_{RL}$ using the gradient

$$\nabla_{\hat{\omega}}\mathcal{L}_{RL}(\hat{\omega}) = \mathbb{E}_\pi\left[(C(\pi) - b_\circ)\nabla_{\hat{\omega}}\log q_\phi(\pi \mid \hat{\omega})\right] \tag{1}$$

where $q_\phi(\pi \mid \hat{\omega}) \equiv \prod_{t=1}^{T} q_\phi(a_t \mid s_t, \hat{\omega})$, and $b_\circ$ is a baseline (we use the baseline proposed in Kwon et al. (2020) for our experiments).

For the second loss $\mathcal{L}_{IL}$, let $\bar{\pi}$ be the best solution found so far for the instance $l$, that consists of the actions $\bar{a}_1, \ldots, \bar{a}_T$. We use teacher forcing to make the decoder $q_\phi(\cdot|s_t, \hat{\omega})$ generate the solution $\bar{\pi}$, during which we obtain the probability values associated with the actions $\bar{a}_1, \ldots, \bar{a}_T$. We increase the log-likelihood of generating $\bar{\pi}$ by adjusting $\hat{\omega}$ using the gradient

$$\nabla_{\hat{\omega}}\mathcal{L}_{IL}(\hat{\omega}) = -\nabla_{\hat{\omega}}\log q_\phi(\bar{\pi} \mid \hat{\omega}) \equiv -\nabla_{\hat{\omega}}\log \prod_{t=1}^{T} q_\phi(\bar{a}_t|s_t, \hat{\omega}). \tag{2}$$

The gradient of the overall loss $\mathcal{L}_{RIL}$ is defined as $\nabla_{\hat{\omega}}\mathcal{L}_{RIL}(\hat{\omega}) = \nabla_{\hat{\omega}}\mathcal{L}_{RL}(\hat{\omega}) + \lambda \cdot \nabla_{\hat{\omega}}\mathcal{L}_{IL}(\hat{\omega})$, where $\lambda$ is a tunable parameter. If a high value for $\lambda$ is selected, the search focuses on generating solutions that are similar to the incumbent solution. This accelerates the convergence of the search policy, which is useful when the number of search iterations is limited.

We note that both decoding processes required for RL and IL can be carried out in parallel, using the same forward pass through the network. Furthermore, only the parameters $\hat{\omega}$ are instance specific, while all other model parameters are identical for all instances. This makes parallelization of multiple instances in a batch more efficient both in time and memory.

### 3.2 ADDED-LAYER UPDATES

We next propose EAS-Lay, which adds an instance-specific residual layer to a trained model. During the search, the weights in the added layer are updated, while the weights of all other original layers are held fixed. We use both RL and IL, similarly to EAS-Emb in Section 3.1.

We formalize EAS-Lay as follows. For each instance $l$ we insert a layer

$$\mathrm{L}^\star(h) = h + ((\mathrm{ReLu}(hW^1 + b^1)W^2 + b^2) \tag{3}$$

into the given decoder $q_\phi$, resulting in a slightly modified model $\tilde{q}_{\phi,\psi}$, where $\psi = \{W^1, b^1, W^2, b^2\}$. The layer takes in the input $h$ and applies two linear transformations with a ReLu activation function in between. The weight matrices $W^1$ and $W^2$ and the bias vectors $b^1$ and $b^2$ are adjusted throughout the search via gradient descent. The weights in the matrix $W^2$ and the vector $b^2$ are initialized to

zero so that the added layer does not affect the output of the model during the first iteration of the search. The gradient for $\mathcal{L}_{RL}$ is given as

$$\nabla_\psi \mathcal{L}_{RL}(\psi) = \mathbb{E}_\pi \left[ (C(\pi) - b_\circ) \nabla_\psi \log \tilde{q}_{\phi,\psi}(\pi) \right], \tag{4}$$

with $\tilde{q}_{\phi,\psi}(\pi) \equiv \prod_{t=1}^T \tilde{q}_{\phi,\psi}(a_t \mid s_t, \omega)$, and $b_\circ$ is a baseline. The gradient for $\mathcal{L}_{IL}$ is defined similarly.

Note that the majority of the network operations are not instance specific. They can be applied identically to all instances running in parallel as a batch, resulting in significantly lower runtime during search. The position at which the new layer is inserted has an impact on the performance of EAS-Lay, and identifying the best position usually requires testing. In general, the memory requirement of this approach can be reduced by inserting the additional layer closer towards the output layer of the network. This decreases the number of layers to be considered during backpropagation. We noticed for transformer-based architectures that applying the residual layer $L^\star(\cdot)$ to the query vector $q$ before it is passed to the single attention head usually results in a good performance.

### 3.3 Tabular updates

EAS-Emb and EAS-Lay require significantly less memory per instance than the original active search. However, they still need to store many gradient weights associated with multiple layers for the purpose of backpropagation. This significantly limits the number of solutions one can generate in parallel. We hence propose EAS-Tab, which does not require backpropagation, but instead uses a simple lookup table to modify the policy of the given model. For each action at a given state, the table provides a guide on how to change its probability, so that the sampled solution has a higher chance at being similar to the best solution found in the past.

Formally, at each step $t$ during the sequential generation of a solution, we redefine the probability of selecting action $a_t$ in the state $s_t$ as $q_\phi(a|s_t, \omega)^\alpha \cdot Q_{g(s_t, a_t)}$ and renormalize over all possible actions using the `softmax` function. Here, $\alpha$ is a hyperparameter, and $g$ is a function that maps each possible state and action pair to an entry in the table $Q$. The network parameters $\theta$ remain unchanged, resulting in fast and memory efficient solution generation. The hyperparameter $\alpha$ is similar to the temperature value proposed in Bello et al. (2016) and modifies the steepness of the probability distribution returned by the model (lower values increase the exploration of the search). During search, the table $Q$ is updated with the objective of increasing the quality of the generated solutions. More precisely, after each iteration, $Q$ is updated based on the best solution $\bar{\pi}$ found so far consisting of the actions $\bar{a}_1, \ldots, \bar{a}_T$ at states $\bar{s}_1, \ldots, \bar{s}_T$, respectively, with

$$Q_{g(s_t, a_t)} = \begin{cases} \max(1, \frac{\sigma}{q_\phi(a|s_t, \omega)^\alpha}), & \text{if } g(s_t, a_t) \in \{g(\bar{a}_1, \bar{s}_1), \ldots, g(\bar{a}_T, \bar{s}_T)\} \\ 1, & \text{otherwise} \end{cases} \tag{5}$$

The hyperparameter $\sigma$ defines the degree of exploitation of the search. If a higher value of $\sigma$ is used, the probabilities for actions that generate the incumbent solution are increased.

In contrast to embedding or added-layer updates, this EAS method requires deeper understanding of the addressed combinatorial optimization problem to design the function $g(s_t, a_t)$. For example, for the TSP with $n$ nodes we use a table $Q$ of size $n \times n$ in which each entry $Q_{i,j}$ corresponds to a directed edge $e_{i,j}$ of the problem instance. The probability increases for the same directed edge that was used in the incumbent solution. This definition of $g(s_t, a_t)$ effectively ignores the information on all the previous visits stored in state $s_t$, focusing instead on the current location (city) in choosing the next move. We note that this EAS approach is similar to the ant colony optimization algorithm (Dorigo et al., 2006), which has been applied to a wide variety of combinatorial optimization problems.

## 4 Experiments

We evaluate all EAS strategies using existing, state-of-the-art RL based methods for three different combinatorial optimization problems. For the first two, the TSP and the CVRP, we implement EAS for the POMO approach (Kwon et al., 2020). For the third problem, the JSSP, we use the L2D method from Zhang et al. (2020). We extend the code made available by the authors of POMO (MIT license) and L2D (no license) with our EAS strategies to ensure a fair evaluation. Note that we only make minor modifications to these methods, and we use the models trained by the authors when available.

We run all experiments on a GPU cluster using a single Nvidia Tesla V100 GPU and a single core of an Intel Xeon 4114 CPU at 2.2 GHz for each experiment. Our source code is available at `https://github.com/ahottung/EAS`. We use the Adam optimizer (Kingma & Ba, 2014) for all EAS approaches. The hyperparameters $\lambda$, $\sigma$, $\alpha$, and the learning rate for the optimizer are tuned via Bayesian optimization using *scikit-optimize* (Head et al., 2020) on separate validation instances, which are sampled from the same distribution as the test instances. The hyperparameters are not adjusted for larger instances used to evaluate the generalization performance.

## 4.1 TSP

The TSP is a well-known routing problem involving finding the shortest tour between a set of $n$ nodes (i.e., cities) that visits each node exactly once and returns to the starting node. We assume that the distance matrix obeys the triangle inequality.

**Implementation** POMO uses a model that is very similar to the AM model from Kool et al. (2019). The model generates instance embeddings only once per instance and does not update them during construction. The probability distribution over all actions are generated by a decoder, whose last layer is a single-headed attention layer. This last layer calculates the compatibility of a query vector $q$ to the key vector $k_i$ for each node $i$. In this operation, the key vector $k_i$ is an embedding that has been computed separately, but identically for each input (i.e., node $i$) during the instance encoding process. For EAS-Emb, we only update the set of single-head keys $k_i$ ($i = 1, \ldots, n$). For EAS-Lay we apply the residual layer $L^\star(\cdot)$ described in Equation 3 to the query vector $q$ before it is passed to the single attention head. For EAS-Tab, we use a table $Q$ of size $n \times n$ and the mapping function $g(s_t, a_t)$ such that each entry $Q_{i,j}$ corespondents to a directed edge $e_{i,j}$ of the problem instance.

**Setup** We use the 10,000 TSP instances with $n = 100$ from Kool et al. (2019) for testing and three additional sets of 1,000 instances to evaluate generalization performance. We evaluate the EAS approaches against just using POMO with greedy action selection, random sampling, and active search as in Bello et al. (2016). In all cases, we use the model trained on instances with $n = 100$ made available by the POMO authors. For greedy action selection, POMO generates $8 \cdot n$ solutions for an instance of size $n$ (using 8 augmentations and $n$ different starting cities). In all other cases, we generate $200 \cdot 8 \cdot n$ solutions per instance (over the course of 200 iterations for the (E)AS approaches). The batch size (the number of instances solved in parallel) is selected for each method individually to fully utilize the available GPU memory. We compare to the exact solver Concorde (Applegate et al., 2006), the heuristic solver LKH3 (Helsgaun, 2017), the graph convolutional neural network with beam search (GCN-BS) from Joshi et al. (2019), the 2-Opt based deep learning (2-Opt-DL) approach from de O. da Costa et al. (2020), the learning improvement heuristics (LIH) method from Wu et al. (2021), the conditional variational autoencoder (CVAE-Opt) approach (Hottung et al., 2021), and deep policy dynamic programming (DPDP) (Kool et al., 2021).

**Results** Table 1 shows the average costs, average gap and the total runtime (wall-clock time) for each instance set. The exact solver Concorde performs best overall, as it is a highly specialized TSP

Table 1: Results for the TSP

| Method | Testing (10k inst.) $n = 100$ | | | Generalization (1k instances) $n = 125$ | | | $n = 150$ | | | $n = 200$ | | |
| | Obj. | Gap | Time | Obj. | Gap | Time | Obj. | Gap | Time | Obj. | Gap | Time |
|---|---|---|---|---|---|---|---|---|---|---|---|---|
| Concorde | 7.765 | 0.000% | 82M | 8.583 | 0.000% | 12M | 9.346 | 0.000% | 17M | 10.687 | 0.000% | 31M |
| LKH3 | 7.765 | 0.000% | 8H | 8.583 | 0.000% | 73M | 9.346 | 0.000% | 99M | 10.687 | 0.000% | 3H |
| GCN-BS | 7.87 | 1.39% | 40M | - | - | - | - | - | - | - | - | - |
| 2-Opt-DL | 7.83 | 0.87% | 41M | - | - | - | - | - | - | - | - | - |
| LIH | 7.87 | 1.42% | 2H | - | - | - | - | - | - | - | - | - |
| CVAE-Opt | - | 0.343% | 6D | 8.646 | 0.736% | 21H | 9.482 | 1.454% | 30H | - | - | - |
| DPDP | 7.765 | **0.004%** | 2H | 8.589 | 0.070% | 31M | 9.434 | 0.942% | 44M | 11.154 | 4.370% | 74M |
| *POMO* Greedy | 7.776 | 0.146% | 1M | 8.607 | 0.278% | <1M | 9.397 | 0.542% | <1M | 10.843 | 1.457% | 1M |
| *POMO* Sampling | 7.770 | 0.074% | 4H | 8.595 | 0.145% | 45M | 9.378 | 0.334% | 78M | 10.838 | 1.416% | 3H |
| *POMO* Active S. | 7.768 | **0.046%** | 5D | 8.591 | 0.095% | 15H | 9.364 | 0.192% | 19H | 10.735 | 0.447% | 24H |
| *POMO* EAS-Emb | 7.769 | 0.063% | 5H | 8.591 | 0.092% | 57M | 9.363 | **0.174%** | 2H | 10.730 | **0.400%** | 4H |
| *POMO* EAS-Lay | 7.769 | 0.053% | 7H | 8.591 | **0.089%** | 74M | 9.363 | 0.176% | 2H | 10.737 | 0.471% | 4H |
| *POMO* EAS-Tab | 7.768 | 0.048% | 5H | 8.591 | 0.091% | 49M | 9.365 | 0.196% | 1H | 10.756 | 0.650% | 3H |

solver. Of the POMO-based approaches, the original active search offers the best gap to optimality, but requires 5 days of runtime. EAS significantly lowers the runtime while the gap is only marginally larger. DPDP performs best among ML-based approaches. However, DPDP relies on a handcrafted and problem-specific beam search, whereas EAS methods are completely problem-independent. On the larger instances, EAS significantly improves generalization performance, reducing the gap over sampling by up to 3.6x. We also evaluate active search using the imitation learning loss, but observe no impact on the search performance (see Appendix B).

## 4.2 CVRP

The goal of the CVRP is to find the shortest routes for a set of vehicles with limited capacity that must deliver goods to a set of $n$ customers. We again use the POMO approach as a basis for our EAS strategies. As is standard in the ML literature, we evaluate all approaches on instance sets where the locations and demands are sampled uniformly at random. Additionally, we consider the more realistic instance sets proposed in Hottung & Tierney (2020) with up to 297 customers (see Appendix A).

**Implementation** We use the same EAS implementation for the CVRP as for the TSP.

**Setup** We use the 10,000 CVRP instances from Kool et al. (2019) for testing and additional sets of 1,000 instances to evaluate the generalization performance. Again, we compare the EAS approaches to POMO using greedy action selection, sampling and active search. We generate the same number of solutions per instance as for the TSP. We compare to LIH, CAVE-Opt, DPDP, NeuRewriter (Chen & Tian, 2019) and neural large neighborhood search (NLNS) from Hottung & Tierney (2020).

**Results** Table 2 shows the average costs, the average gap to LKH3 and the total wall-clock time for all instance sets. EAS-Lay outperforms all other approaches on the test instances, including approaches that rely on problem-specific knowledge, with a gap that beats LKH3. Both other EAS methods also find solutions of better quality than LKH3, which is quite an accomplishment given the many years of work on the LKH3 approach. We note it is difficult to provide a fair comparison between a single-core, CPU-bound technique like LKH3 and our approaches that use a GPU. Nonetheless, assuming a linear speedup, at least 18 CPU cores would be needed for LKH3 to match the runtime of EAS-Tab. On the generalization instance sets with $n = 125$ and $n = 150$, the EAS approaches also outperform LKH3 and CVAE-Opt while being significantly faster than active search. On the instances with $n = 200$, active search finds the best solutions of all POMO based approaches with a gap of 0.22% to LKH3, albeit with a long runtime of 36 hours. We hypothesize that significant changes to the learned policy are necessary to generate high-quality solutions for instances that are very different to those seen during training. Active search's ability to modify all model parameters makes it easier to make those changes. EAS-Tab offers the worst performance on the instances with $n = 200$ with a gap of 11.8%. This is because EAS-Tab is very sensitive to the selection of the hyperparameter $\alpha$, meaning that EAS-Tab requires hyperparameter tuning on some problems to generalize more effectively. Adjusting $\alpha$ for the $n = 200$ case improves EAS-Tab's gap to at least 3.54%, making it slightly better than greedy or sampling.

Table 2: Results for the CVRP on instances with uniformly sampled locations and demands

| Method | **Testing** (10k inst.) $n = 100$ | | | **Generalization** (1k instances) $n = 125$ | | | $n = 150$ | | | $n = 200$ | | |
| | Obj. | Gap | Time | Obj. | Gap | Time | Obj. | Gap | Time | Obj. | Gap | Time |
|---|---|---|---|---|---|---|---|---|---|---|---|---|
| LKH3 | 15.65 | 0.00% | 6D | 17.50 | 0.00% | 19H | 19.22 | 0.00% | 20H | 22.00 | 0.00% | 25H |
| NLNS | 15.99 | 2.23% | 62M | 18.07 | 3.23% | 9M | 19.96 | 3.86% | 12M | 23.02 | 4.66% | 24M |
| NeuRewriter | 16.10 | - | 66M | - | - | - | - | - | - | - | - | - |
| LIH | 16.03 | 2.47% | 5H | - | - | - | - | - | - | - | - | - |
| CVAE-Opt | - | 1.36% | 11D | 17.87 | 2.08% | 36H | 19.84 | 3.24% | 46H | - | - | - |
| DPDP | 15.63 | **-0.13%** | 23H | 17.51 | 0.07% | 3H | 19.31 | 0.48% | 5H | 22.26 | 1.20% | 9H |
| Greedy | 15.76 | 0.76% | 2M | 17.73 | 1.29% | <1M | 19.64 | 2.18% | 1M | 22.90 | 4.12% | 1M |
| Sampling | 15.67 | 0.17% | 7H | 17.60 | 0.54% | 73M | 19.48 | 1.35% | 2H | 23.18 | 5.35% | 5H |
| Active S. | 15.63 | -0.07% | 8D | 17.47 | -0.21% | 25H | 19.21 | -0.03% | 29H | 22.05 | **0.22%** | 36H |
| EAS-Emb | 15.63 | -0.08% | 9H | 17.47 | -0.21% | 93M | 19.22 | 0.03% | 3H | 22.19 | 0.88% | 6H |
| EAS-Lay | 15.61 | **-0.23%** | 12H | 17.46 | **-0.24%** | 2H | 19.21 | **-0.04%** | 3H | 22.10 | 0.45% | 8H |
| EAS-Tab | 15.62 | -0.14% | 8H | 17.50 | 0.00% | 80M | 19.36 | 0.72% | 2H | 24.56 | 11.8% | 5H |

### 4.3 JSSP

The JSSP is a scheduling problem involving assigning jobs to a set of heterogeneous machines. Each job consists of multiple operations that are run sequentially on the set of machines. The objective is to minimize the time needed to complete all jobs, called the makespan. We evaluate EAS using the L2D approach, which is a state-of-the-art ML based construction method using a graph neural network.

**Implementation** L2D represents JSSP instances as disjunctive graphs in which each operation of an instance is represented by a node in the graph. To create a schedule, L2D sequentially selects the operation that should be scheduled next. To this end, an embedding $h_v$ is created for each node $v$ in a step-wise encoding process. In contrast to POMO, the embeddings $h_v$ are recomputed after each decision step $t$. Since EAS-Emb requires static embeddings, we modify the network to use $\tilde{h}_v^t = h_v^t + h_v^{ST}$ as an embedding for node $v$ at step $t$, where $h_v^{ST}$ is a vector that is initialized with all weights being set to zero. During the search with EAS-Emb we only adjust the static component $h_v^{ST}$ of the embedding with gradient descent. For EAS-Lay, we insert the residual layer $L^\star(\cdot)$ described in Equation 3 to each embedding $h_v$ separately and identically. Finally, for EAS-Tab, we use a table $Q$ of size $|O| \times |O|$, where $|O|$ is the number of operations, and we design the function $g(s_t, a_t)$ so that the entry $Q_{i,j}$ corresponds to selecting the operation $o_j$ directly after the operation $o_i$.

**Setup** We use three instance sets with 100 instances from Zhang et al. (2020) for testing and to evaluate the generalization performance. We use the exact solver Google OR-Tools (Perron & Furnon) as a baseline, allowing it a maximum runtime of 1 hour per instance. Furthermore, we compare to L2D with greedy action selection. Note that the performance of the L2D implementation is CPU bound and does not allow different instances to be batch processed. We hence solve instances sequentially and generate significantly fewer solutions per instance than for the TSP and the CVRP. For sampling, active search and the EAS approaches we sample 8,000 solutions per problem instance over the course of 200 iterations for the (efficient) active search approaches.

**Results** Table 3 shows the average gap to the OR-Tools solution and the total wall-clock time per instance set. EAS-Emb offers the best performance for all three instance sets. On the $10 \times 10$ instances, EAS-Emb reduces the gap by 50% in comparison to pure sampling. Even on the $20 \times 15$ instances it reduces the gap to 16.8% from 20.8% for pure sampling, despite the low number of sampled solutions per instance. EAS-Lay offers performance that is comparable to active search. We note that if L2D were to more heavily use the GPU, instances could be solved in batches, thus drastically reducing the runtime of EAS-Lay and EAS-Tab. While EAS-Tab shows similar performance to active search on the test instance set, it is unable to generalize effectively to the larger instances.

### 4.4 SEARCH TRAJECTORY ANALYSIS

To get a better understanding of how efficient active search improves performance, we monitor the quality of the solutions sampled at each of the 200 iterations of the search. Figure 2 reports the average quality over all test instances for the JSSP and over the first 1,000 test instances for the TSP and CVRP. As expected, the quality of solutions generated via pure sampling does not change over the course of the search for all three problems. For all other methods, the quality of the generated solutions improves throughout the search. Thus, all active search variants successfully modify the (model) parameters in a way that increases the likelihood of generating high-quality solutions.

Table 3: Results for the JSSP

| | Testing (100 inst.) | | | Generalization (100 instances) | | | | | |
| | $10 \times 10$ | | | $15 \times 15$ | | | $20 \times 15$ | | |
| Method | Obj. | Gap | Time | Obj. | Gap | Time | Obj. | Gap | Time |
|---|---|---|---|---|---|---|---|---|---|
| OR-Tools | 807.6 | 0.0% | 37S | 1188.0 | 0.0% | 3H | 1345.5 | 0.0% | 80H |
| Greedy | 988.6 | 22.3% | 20S | 1528.3 | 28.6% | 44S | 1738.0 | 29.2% | 60S |
| Sampling | 871.7 | 8.0% | 8H | 1378.3 | 16.0% | 25H | 1624.6 | 20.8% | 40H |
| Active S. | 854.2 | 5.8% | 8H | 1345.2 | 13.2% | 32H | 1576.5 | 17.2% | 50H |
| EAS-Emb | 837.0 | **3.7%** | 7H | 1326.4 | **11.7%** | 22H | 1570.8 | **16.8%** | 37H |
| EAS-Lay | 859.6 | 6.5% | 7H | 1352.6 | 13.8% | 25H | 1581.8 | 17.6% | 46H |
| EAS-Tab | 860.2 | 6.5% | 8H | 1376.8 | 15.9% | 29H | 1623.4 | 20.7% | 51H |

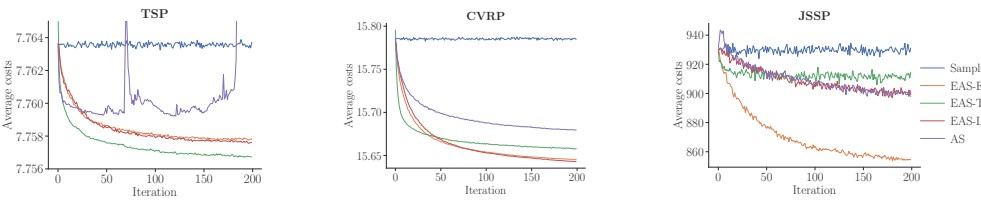

Figure 2: Average costs of sampled solutions at each iteration (best viewed in color).

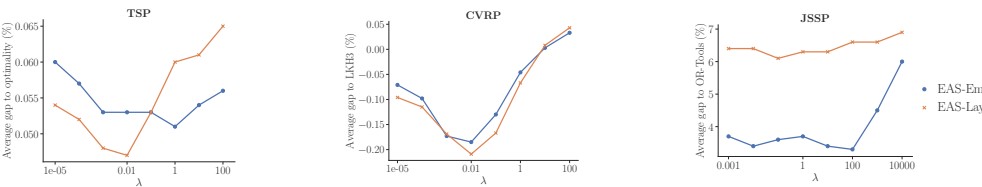

Figure 3: Influence of $\lambda$ on the solution quality for EAS-Emb and EAS-Lay.

For the TSP, EAS-Emb and EAS-Lay offer nearly identical performance, with EAS-Tab outperforming both by a very slight margin. The original active search is significantly more unstable, which is likely the result of the learning rate being too high. Note that the learning rate has been tuned on an independent validation set. These results indicate that selecting a suitable learning rate is significantly more difficult for the original active search than for our efficient active search variants where only a subset of (model) parameters are changed. For the CVRP, all EAS variants find better solutions on average than the original search after only a few iterations. Keeping most parameters fixed seems to simplify the underlying learning problem and allows for faster convergence. For the JSSP, EAS-Emb offers significantly better performance than all other methods. The reason for this is that the L2D approach uses only two node features and has a complex node embedding generation procedure. While the original active search must fine tune the entire embedding generation process to modify the generated solutions, EAS-Emb can just modify the node embedding directly.

### 4.5 ABLATION STUDY: IMITATION LEARNING LOSS

We evaluate the impact of the imitation learning loss $\mathcal{L}_{IL}$ of EAS-Emb and EAS-Lay with a sensitivity and ablation analysis for the hyperparameter $\lambda$. We solve the first 500 test instances (to reduce the computational costs) for the TSP and CVRP, and all test instances for the JSSP using EAS-Emb and EAS-Lay with different $\lambda$ values. The learning rate remains fixed to a value determined in independent tuning runs in which $\lambda$ is fixed to zero. Figure 3 shows the results for all three problems. For the TSP and the CVRP, the results show that $\mathcal{L}_{IL}$ can significantly improve performance. When $\lambda$ is set to 0 or very small values, $\mathcal{L}_{IL}$ is disabled, thus including $\mathcal{L}_{IL}$ is clearly beneficial on the TSP and CVRP. For the JSSP, the inclusion of $\mathcal{L}_{IL}$ does not greatly improve performance, but it does not hurt it, either. Naturally, $\lambda$ should not be selected too low or too high as either too little or too much intensification can hurt search performance.

## 5 CONCLUSION

We presented a simple technique that can be used to extend ML-based construction heuristics by an extensive search. Our proposed modification of active search fine tunes a small subset of (model) parameters to a single instance at test time. We evaluate three example implementations of EAS that all result in significantly improved model performance in both testing and generalization experiments on three different, difficult combinatorial optimization problems. Our approach of course comes with some key limitations. Search requires time, thus for applications needing extremely fast (or practically instant) solutions, greedy construction remains a better option. Furthermore, while the problems we experiment on have the same computational complexity as real-world optimization problems, additional work may be needed to handle complex side constraints as often seen in industrial problems.

ACKNOWLEDGMENTS

The computational experiments in this work have been performed using the Bielefeld GPU Cluster.

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

## A   Experiments for more realistic CVRP instances

We provide results from additional experiments for the CVRP on more realistic instances to show that our approach is effective at solving instances with a wide range of structures. Our EAS methods are implemented in the same way as in Section 4.2

**Setup** We evaluate EAS on 9 instance sets from Hottung & Tierney (2020) (consisting of 20 instances each) that have been generated based on the instances from Uchoa et al. (2017) performing 3 runs per instance. The characteristics of the instances vary significantly between sets, but all instances in the same set have been sampled from an identical distribution. For each instance set we train a new model for 3 weeks on a separate, corresponding training set. For testing, we run all (efficient) active search approaches for 200 iterations using the newly trained models. Additionally, we test the generalization performance by solving all instance sets with EAS-Lay using the CVRP model of Section 4.2 that has been trained on the uniform instances (with $n = 100$) from Kool et al. (2019) and call this Lay*. We only evaluate the generalization performance of EAS-Lay (the best performing EAS approach from Section 4.2) to keep the computational costs low. In all experiments, we use hyperparameters tuned for the uniform CVRP instances. We compare to NLNS, LKH3 and the state-of-the-art unified hybrid genetic search (GS) from Vidal et al. (2014). As is standard in the operations research literature, we round the distances between customers to the nearest integer. Furthermore, we solve instances sequentially and not in batches of different instances. However, to make better use of the available GPU memory, we solve up to 10 copies of the same instance in parallel for the EAS approaches and for POMO with sampling. The best solution found so far is shared between all runs, which has an impact on the imitation learning loss $\mathcal{L}_{IL}$ for EAS-Emb and EAS-Lay. For EAS-Tab we set $\tilde{Q} = (1 - \beta) \cdot Q + \beta \cdot Q^{glob}$, where $Q^{glob}$ is the lookup table for the best solution over all runs and $\beta$ is linearly increased from 0 to 1 over the course of the search.

**Results** Table 4 shows the gap to the unified hybrid genetic search and the average runtime per instance for all methods. For EAS-Lay we report the performance of the instance set specific models and additionally the generalization performance when using the model trained on uniform CVRP instances (with $n = 100$). The later results are marked with a star. EAS-Emb and EAS-Lay both find better solution than NLNS and LKH3 on 8 out of the 9 instance sets. EAS-Tab outperforms NLNS and LKH3 on all but two instance sets. As a side note, we have found that the original active search (AS) performs surprisingly well, outperforming LKH3 on 3 instance sets, even though it still cannot surpass our newly proposed EAS methods. The version of EAS-Lay (marked with a star) that uses the model trained on uniform instances with $n = 100$ performs surprisingly well with gaps between 0.26% to 4.08% to the GS.

Table 4: Results for the CVRP on the instance sets from Hottung & Tierney (2020).

| | | Gap to GS in % | | | | | | | Avg. Runtime in minutes | | | | | | | | |
| | | POMO | | POMO-EAS | | | | | | POMO | | POMO-EAS | | | | | | |
| Inst. | $n$ | Sam. | AS | Emb | Lay | Lay* | Tab | NLNS | LKH | Sam. | AS | Emb | Lay | Lay* | Tab | NLNS | LKH | GS |
|---|---|---|---|---|---|---|---|---|---|---|---|---|---|---|---|---|---|---|
| XE_1 | 100 | 0.86 | 0.65 | **0.23** | 0.26 | 0.61 | 0.31 | 0.32 | 2.12 | 0.9 | 1.3 | 1.3 | 1.4 | 1.5 | 0.9 | 3.2 | 6.2 | 0.6 |
| XE_3 | 128 | 0.76 | 0.80 | 0.26 | **0.25** | 0.26 | 0.29 | 0.44 | 0.54 | 1.3 | 1.6 | 1.8 | 2.0 | 2.1 | 1.4 | 3.2 | 2.0 | 1.2 |
| XE_5 | 180 | 0.51 | 0.20 | **0.09** | **0.09** | 0.54 | 0.13 | 0.58 | 0.16 | 2.5 | 2.2 | 3.3 | 3.7 | 3.8 | 2.7 | 3.2 | 1.1 | 1.4 |
| XE_7 | 199 | 1.50 | 0.88 | **0.37** | 0.45 | 1.29 | 0.80 | 2.03 | 0.72 | 3.3 | 2.4 | 4.2 | 4.7 | 4.9 | 3.5 | 3.2 | 3.6 | 2.4 |
| XE_9 | 213 | 1.96 | 1.30 | **0.64** | 0.71 | 4.08 | 0.83 | 2.26 | 1.09 | 3.7 | 2.5 | 4.6 | 5.2 | 5.4 | 3.9 | 10.2 | 1.1 | 2.4 |
| XE_11 | 236 | 1.42 | 1.22 | 0.82 | 0.84 | 1.76 | 0.94 | **0.65** | 0.78 | 4.0 | 2.7 | 5.2 | 5.1 | 5.6 | 4.8 | 10.2 | 1.1 | 3.2 |
| XE_13 | 269 | 1.40 | 0.88 | **0.38** | 0.56 | 2.83 | 0.80 | 0.82 | 1.55 | 7.1 | 3.5 | 8.7 | 6.6 | 7.1 | 7.6 | 10.2 | 5.7 | 3.6 |
| XE_15 | 279 | 1.81 | 2.17 | **0.85** | 0.96 | 2.51 | 1.27 | 1.81 | 1.32 | 7.3 | 3.3 | 8.8 | 6.6 | 7.2 | 7.9 | 10.3 | 5.8 | 5.5 |
| XE_17 | 297 | 1.66 | 0.97 | **0.44** | 0.65 | 2.15 | 0.92 | 1.41 | 1.23 | 8.9 | 3.8 | 8.8 | 6.8 | 7.1 | 9.4 | 10.3 | 2.5 | 4.2 |

## B   Ablation Study: Active search loss

We evaluate if applying the imitation learning loss component used by EAS-Emb and EAS-Lay to the original active search can significantly improve the performance. To this end, we solve all test instances using active search with and without the imitation learning loss component. Note that the hyperparameters for each approach have been tuned independently on separate validation set instances. Table 5 shows the results. We observe no significant impact of the imitation learning loss $\mathcal{L}_{IL}$ on the performance of active search. This means that active search with imitation learning loss is

Table 5: Performance of active search with and without imitation learning loss component.

| Active search loss | TSP (10k inst.) | | | CVRP (10k inst.) | | | JSSP (100 inst.) | | |
|---|---|---|---|---|---|---|---|---|---|
| | Obj. | Gap | Time | Obj. | Gap | Time | Obj. | Gap | Time |
| RL | 7.768 | 0.046% | 5D | 15.634 | -0.070% | 8D | 854.20 | 5.80% | 8H |
| RL and IL | 7.769 | 0.052% | 5D | 15.634 | -0.073% | 8D | 854.16 | 5.78% | 9H |

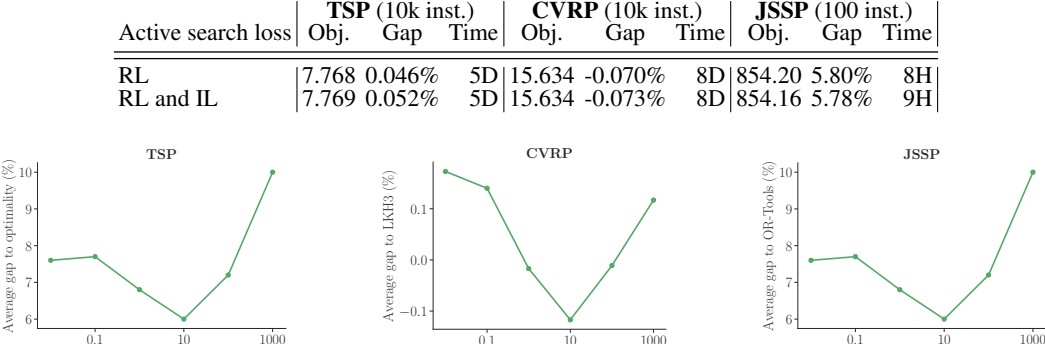

Figure 4: Influence of $\sigma$ on the solution quality for EAS-Tab

not competitive with EAS-Lay and EAS-Emb across all problems, even when sharing the same loss function.

## C   PARAMETER SWEEP: EAS-TAB INTENSIFICATION

We investigate the impact of the hyperparameter $\sigma$ on EAS-Tab, which controls the degree of exploitation of the search. By setting $\sigma$ to zero (or very small values) we essentially disable the lookup table, thus examining its impact on the search. We again solve all three problems with different values of $\sigma$ on a subset of test instances. We fix $\alpha$ independently based on tuning on a separate set of validation instances. Figure 4 provides the results for adjusting $\sigma$. For all three problems, $\sigma = 10$ provides the best trade-off between exploration and exploitation. Note that low $\sigma$ values (which reduce the impact of the lookup table updates) hurt performance, meaning that the table based adjustments are effective in all cases.

