# OpenReview forum: "Efficient Active Search for Combinatorial Optimization Problems"
_ICLR.cc/2022/Conference — ICLR 2022 Poster_

### Official Review · Reviewer_4SkE · 2021-10-24

**Correctness:** 4
**Technical Novelty And Significance:** 3
**Empirical Novelty And Significance:** 3
**Recommendation:** 6
**Confidence:** 3

**Main Review:**

Strengths:
- This paper studies an exciting area—machine learning for combinatorial optimization—where machine learning has the potential to make a big impact.
- From the experiments (especially Tables 1 and 2), it looks as through the proposed approaches are much faster than competitor, active search [Bello et al. ‘16], which (from my understanding) searches for ways to adjust all parameters of the trained model at test time. In contrast, the proposed approach only searches for ways to adjust specific subsets of model parameters, which makes the approach faster.
- I appreciate that the authors evaluate their approach on a few different types of combinatorial optimization problems: two different types of routing problems and a scheduling problem. For the scheduling problem, the improvement over active search is a bit more modest.

Weaknesses:
- I found the problem description somewhat hard to follow. In Section 3, it would be helpful to clarify what exactly an “action” corresponds to in this setting. One way to do this would be to summarize the combinatorial problems studied in the experiments section and explain what an action corresponds to and what the state $s_{t+1}$ corresponds to after applying an action $a_t$.
- In terms of solution quality, the improvements over problem-specific baselines are sometimes really small (e.g., in Tables 1 and 2, a fraction of a percentage). On such a small scale, I wasn’t sure if I could trust the superiority of any particular method. Confidence intervals would really help here.

Detailed comments:
- Page 2: I’m not sure that “exemplary” is the right word here; I would remove it.
- Equation (1): I’m not sure what you mean when you say that $b_0$ is a baseline used to reduce variance. Can you say more?

**Summary Of The Paper:**

This paper studies deep learning methods for solving combinatorial optimization problems. The authors write that state-of-the-art methods typically use models that consist of encoder and decoder units. The methods first create an embedding of the problem instance using the encoder. Then, starting with an empty solution to the problem, the embedding and the decoder are used to autoregressively construct a solution over a series of time steps. Given an already trained model and a test instance, this paper studies how to quickly update the model parameters in order to improve the quality of the solution returned by this procedure. The authors propose three techniques, which adjust (1) the normally static embeddings of the problem instance that are generated by the encoder model, (2) the weights of additional instance-specific residual layers added to the decoder, and (3) the parameters of a lookup table that directly affect the probability distribution returned by model.

**Summary Of The Review:**

Overall, I’m leaning toward acceptance because the proposed approach seems to provide a notable improvement over prior methods (in particular, active search by Bello et al. ‘16]) in terms of runtime.

---

> ### Author Response · Authors · 2021-11-22
> **Response**
>
> Thank you for reviewing our paper and for your positive feedback! We are happy that you find the contribution of our paper to be significant and that you like our evaluation on different problems.
>
> Please let us address your stated weaknesses and comments:
>
> **Weaknesses**
>
> *  We slightly modified that paragraph and now state that  “for the TSP, each action corresponds to visiting a different city next”. Other components (e.g., the starting state) are also explained using the TSP. Please see the updated version of the paper. We hope that this change makes this paragraph easier to understand. We would really like to extend this even more. However, we are struggling to stay within the given page limit, and we do not know which other parts of the paper we could shorten.
> * Please note that for Table 1 and 2, the considered instance sets are very large (10,000 and 1,000 instances). Furthermore, all methods are evaluated on exactly the same instances (per problem size). In other words, instead of generating instances on the fly, we pre-compute instance sets (one per problem size) and use these in our experiments. We think that confidence intervals are not a proper tool to compare algorithms in that setting (and they are usually not reported in related work). For the JSSP (Table 3), the considered instance set is smaller (100 instances). However, the results are much more clear-cut with larger relative gaps between the methods.
>
> **Detailed comments**
>
> * *Page 2*: We removed that word.
> * *Equation (1)*: Yes, we were too vague. We now state that we use the baseline proposed in Kwon et al. (2020).

---

### Official Review · Reviewer_xF8a · 2021-11-02

**Correctness:** 4
**Technical Novelty And Significance:** 2
**Empirical Novelty And Significance:** 3
**Recommendation:** 6
**Confidence:** 3

**Main Review:**

Significance: One limitation of existing RL-based approaches for combinatorial optimization is its resource requirement. As demonstrated in Table 1, the active search technique in Bello et al. (2016) takes 5 days to solve 10,000 test instances of TSP. The paper aims to tackle this limitation by proposing to optimize only a subset of the model parameters. I think the paper is making a good and meaningful contribution towards research in the field.

Novelty: The paper extends the active search method in Bello et al. (2016). The three proposed implementations are based on one general idea of optimizing only a subset of the model parameters. The novelty of the proposed technique is therefore limited. However, if the method performs well, its simplicity could be of high interest.

Presentation: The paper is well-written. The related literature is discussed in detail. The experimental results are clearly presented, with ablation study and trajectory analysis. There are some minor ambiguities in presenting the proposed techniques, as elaborated below.

There are some ambiguities in the paper:

1. On page 4, below figure 1, the paper proposes the first strategy: update the embedding of the using the loss function J_{RL} and J_{IL}. These loss functions are not explicitly specified anywhere in the paper. Only their gradient w.r.t. the embeddings are presented in Eq.(1) and Eq.(2). Readers who are not familiar with RL/Imitation Learning literature may not know what J_{RL} and J_{IL} are. It would be great if the author can be more explicit about the loss functions before presenting their gradients.

2. In Table 1, the authors provide wall-clock time for the proposed algorithms and other baselines on a set of 10,000 TSP instances. EAS achieves competitive performance while taking only 5-7 hours to run, as compared to 5 days using the original active search. Is this improvement due to:
(a) EAS uses less memory, and hence we can solve more instances in parallel, or
(b) EAS is computationally more efficient, i.e., it uses less CPU-time to achieve the competitive performance, or
(c) a combination of the above?
It would be better if the authors report the CPU-time (instead of wall-clock time), and separately report the space (memory) and time (CPU) of these algorithms.

Minor comments:
In figure 3, the x-axis should be labeled lambda.


**Summary Of The Paper:**

The paper studies machine learning-based methods for combinatorial optimization. The paper builds upon Bello et al. (2016) on using reinforcement learning to generate solutions for combinatorial optimization problems (e.g., TSP). The novelty of the paper is to optimize only a subset of the model parameters. The paper then proposes three different implementations based on this idea.

**Summary Of The Review:**

The paper proposes a simple extension to an existing RL-based method for combinatorial optimization. Its effectiveness is demonstrated empirically. However, I feel that the results of the experiments should be reported in greater detail, i.e., to compare with the original active search in different performance metrics such as memory and CPU time usage.

---

> ### Author Response · Authors · 2021-11-22
> **Response**
>
> Thank you for reviewing our paper and for your positive feedback! We are happy that you think that our paper makes a meaningful contribution to the field and that you find it to be well written.
>
> Please let us address your two remaining concerns/comments:
>
> 1. You are right that this might be confusing for some readers. We updated the paper and now describe the two loss functions more detailed at the beginning of Section 3.1. Furthermore, we changed the notation from $J_{RL}$ to $\mathcal{L}\_{RL}$ to make it more clear that  $\mathcal{L}\_{RL}$ is a loss function that should be minimized. We still do not explicitly state the full loss functions in all cases, because we would need to significantly cut other parts of the paper to fit in the required 4 equations. If you believe that the loss functions are still difficult to understand, we could also provide them explicitly in the Appendix of the camera-ready version.
> 2. EAS is much faster because it uses less memory per instance and is more efficient (so answer c). The low memory consumption per instance allows EAS to solve multiple instances in parallel. Furthermore, even if we solve a single instance at a time via EAS, it is faster than the original active search. This is because significantly fewer weights need to be updated during the search.
> Reporting the runtime and memory usage of GPU-based algorithms is somewhat difficult.  We believe that the fairest way is to show the total wall-clock time while fully utilizing the available GPU memory. This single number is much easier to interpret than CPU-time, GPU-time, CPU-memory and CPU-time. The CPU-time alone would be much lower than the actual time that it takes to solve an instance (since most operations are performed on a GPU).
>
> *Minor comments*: We have added lambda to the x-axis label.

---

> > ### Comment · Reviewer_xF8a · 2021-11-29
> > **Follow-up Response**
> >
> > Dear authors,
> >
> > Thank you for your response and for updating the paper. I am generally happy with the response. However, I still think that the evaluation metric (e.g., wall-clock time) is relatively limited (as mentioned in the original reviews). The novelty and contribution of the paper are good but not exceptional.
> >
> > For these reasons, I'd like to keep my original score.

---

### Official Review · Reviewer_cAC5 · 2021-11-02

**Correctness:** 4
**Technical Novelty And Significance:** 3
**Empirical Novelty And Significance:** 2
**Recommendation:** 8
**Confidence:** 4

**Main Review:**

**Strengths**

1. The paper is clear, well organised and well written
1. The presented approach seems applicable to any constructive method as long as it has a encoder/decoder type of architecture
1. In the experiments, the proposed approach is applied to 2 models for solving 3 different problems and the results are consistently positive, which hints at the generality of the proposed approach.
1. It improves the performance of the underlying model on test instances from the same distribution as the training instances as well as to larger instances (from the same distribution), effectively addressing the well-known difficulty of standard learning-based models to perform well on larger instances
1.Nice discussion in Sec 4.4. to explain possible reasons of why one of the proposed variants work best for each problem.

**Weaknesses**
1. In the experiments, the scale of instances is limited: 200 nodes for TSP and CVRP, while recent learning-based methods such as the cited [Fu et al 2021] manage to solve TSP instances with up to 10,000 nodes.
1. Generalisation is only illustrated (and claimed) with respect to the size of the instances. It would be interesting to see the results on other distributions shifts (e.g. applying EAS to a model trained on TSP100 on instances of TSPlib)
1. For CVRP, results are indeed provided for other distributions. But for each family of instances, the authors say that the model is trained for 3 weeks and then tested on similar instances for each family. Could EAS be helpful to learn good solutions starting from the same model?


**Recommendation**

I would vote for accepting the paper. The contribution is interesting as a middle ground between fine-tuning a whole model for each instance (active search) and other non-learning based search strategies (beam-search, sampling, etc). The proposed approach is illustrated on 2 models for 3 standard CO problems and consistently shows good results.

**Questions**
1. What is the motivation of adding a new layer to fine-tune (EAS-Lay) versus fine-tuning some existing layers of the model?
1. In Tables 1 and 2, why are there no entries for most of the learning-based baselines for N >= 125 ? Since there is no strict time-limit in this setting, I guess with a small-enough batch size for the models to fit into memory, all these methods would provide some results for N up to 200.
1. In Appendix Table 4, have you tried simply applying EAS to the model trained on the uniform distribution CVRP100? That would be a natural test of the  impact of EAS on generalization.
1. Looking at Figure 3,  it seems the value of the best lambda depends on the problem and the range of potential values is quite wide (0,01-100). Have you checked the scale of the different losses and could it help explain such a difference?
1. If one were to apply EAS to another model/CO problem, could you deduce from your experiments a general kind of rule of thumb of which variant would work better for which situation?


**Additional feedback**
* In Introduction
    *  “these methods do not react towards the solutions seen so far, i.e., the underlying distribution from which instances are sampled is never changed throughout the search”. Not clear to me. Do you mean …from which solutions are sampled?
    * “..wide adaption” —> adoption
* Sec 3.Background: the decoder is introduced with parameter $\omega$, but this one is only defined as the embeddings in the next paragraph
* Sec 3.1: In the definition of the total gradient right after equation (2), shouldn’t there be a minus before one of the gradients?  Since one gradient corresponds to the minimisation of the cost and the other to the maximisation of the likelihood.
* Figure 2: y-axis is the average costs. Optimality gap would be more relevant (and consistent) especially if instances have different sizes
*




**Summary Of The Paper:**

The paper deals with end-to-end learning of heuristics for combinatorial optimization problems. The authors propose an extension of the active search method of [Bello et al 2016], where only part of the model parameters are updated at test time for each instance. They propose three ways of applying this idea that consist in fine-tuning part of the instance embeddings, the parameters of an additional layer or directly the prediction scores of the model. Applied to the POMO method [Kwon et al 2020] for the TSP and CVPR and the L2D method of [Zhang et al 2020] for the JSSP, the proposed efficient active search leads to significant improvements on instances of the same size and larger than the training ones.


**Summary Of The Review:**

The paper provides an interesting contribution to learn to search for high-quality solutions at test time, nicely completing end-to-end learning pipelines for solving CO problems. The proposed approach could be applied to any model that has an encoder-decoder type of architecture, and is experimentally validated on 2 models and 3 problems. A limitation is that it is not clear if it could help a model generalize to instances that are much larger than the training ones, or with significantly different characteristics. I vote for accepting the paper.


### Update after rebuttal

I thank the authors for precisely answering all my questions and concerns.
I am happy to confirm my initial recommendation of accepting the paper.

---

> ### Author Response · Authors · 2021-11-22
> **Response**
>
> Thank you for reviewing our paper and for your very positive feedback! We are very happy that you find our method/contribution interesting and that you believe that it can be successfully applied to a wide variety of problems. Please let us address your valuable comments!
>
> **Weaknesses**
>
> 1. *Instance size*: You are right that there exist some works that focus on solving much larger instances. However, the considered RL-based methods that build solutions sequentially (i.e., POMO and L2D) are currently not well suited to tackle large scale instances (e.g., instances with 10,000 nodes). However, while TSP instances with 100 nodes are trivial to solve (see the performance of Concorde in Table 1), the CVRP and JSSP instance sizes in our paper are difficult very difficult to solve, despite not containing many nodes (see the performance of LKH3 in Table 2).  The CVRP with less than 100 customers is still used to evaluate state-of-the-art operations research techniques. In fact,  40% of the instances contained in the CVRPLIB have less than 100 customers.
> 2. & 3. *Generalization*: You ask for some more insights on how well EAS generalizes to distribution shifts (in contrast to the generalization to larger instances, e.g., instances with more customers). We follow your great suggestion in Question 3, and we will use the CVRP100 model to solve the different real-world based XE instances using EAS-Lay. This allows us to evaluate the generalization performance of EAS to instances that have been sampled from a vastly different distribution. We propose to only report results for EAS-Lay to keep the computational costs of the experiment low (we think that it can be reasonably assumed that all EAS approaches perform somewhat similar). However, if you believe that results for EAS-Emb and EAS-Tab are also of interest, we will also run those experiments. We will report the results in the Appendix of the camera-ready version.
>
> **Questions**
>
> 1. We added an extra layer because it increases the capacity of the model. This could theoretically improve the performance of our approach. Furthermore, the size of that new layer can be adjusted independent of the original model architecture.  That being said, we believe that adjusting an existing layer should lead to very similar results to our EAS variant.
> 2. For the ML-based algorithms, we currently report the results obtained by the respective authors. In most cases, generalization results of already trained models are not available. Performing a full training run for each method would be very time intensive. However, you are of course right that more results would be interesting here. Luckily, the source code of DPDP and NLNS is now available (including model checkpoints). We will run both methods on our generalization datasets and report the results in the camera-ready version.
> 3. See our comment regarding Weakness 2 & 3. We will run the experiments and include the results in the paper. Thank you for suggesting this.
> 4. Yes. Especially, the scale of the imitation learning loss varies between problems. However, we believe that this is not a big concern since the lambda parameter needs to be tuned anyway. Future research could investigate if it is possible to reduce the range of sensible lambda values by scaling the imitation learning loss dynamically.
> 5. This is a good question, but answering it is not easy because there are so many factors at play. We don’t think that we could reliably predict which of our methods would work best on a new problem. However, implementing EAS-Lay and EAS-Emb is very easy (meaning that if you implemented an RL-based approach it might take you about 30 minutes to implement both EAS methods). Furthermore, evaluating EAS is also cheap (no model training is needed; a validation set size of around 100 instances should be enough for first results, meaning that you only need 5 GPU hours for evaluation per method). So we would recommend implementing both of these two methods and to see which one works best. We would recommend EAS-Tab only in case that it is straightforward to design the function $g(s_t, a_t)$.
>
> **Additional feedback:**
>
> * *Introduction*: We fixed both errors in the updated version of the paper. Thanks!
> * *Background paragraph*: $\omega$ is already introduced in the third sentence of the paragraph. Or are we missing something?
> * *Total gradient*: Yes, you are right. We have added a minus to Equation 2. Thank your for pointing this out.
> * *Figure 2*: You are right that it would be better to show the gap to optimality. However, we do not have the optimal solutions for the CVRP and JSSP instances. Note that the instance sizes are identical for each problem (100 nodes for TSP and CVRP, 10x10 for JSSP).

---

### Official Review · Reviewer_JA4x · 2021-11-02

**Correctness:** 4
**Technical Novelty And Significance:** 3
**Empirical Novelty And Significance:** 3
**Recommendation:** 8
**Confidence:** 4

**Main Review:**

The paper presents an interesting idea that seems to have a large impact. The
evaluation is thorough and fair, the results are convincing. This is a good
paper that should be accepted.

There are a few minor points that were unclear to me and might warrant further
discussion. The results for the TSP in Table 1 show that concorde is often the
fastest solver. This is somewhat counter-intuitive, especially compared to LKH,
as it is a complete solver. The proposed method is also often much slower. Some
explanation of this would be helpful for the reader to understand what exactly
is going on there.

The most nebulous part of the proposed method to me is the placement of the new
layer, which sounds like it might be quite difficult in practice and potentially
require expensive evaluation of different alternatives. A more in-depth
discussion of how the authors determined this for their experiments, along with
some recommendations on how to do this in a new setting, would make the paper
stronger and more applicable in practice.


**Summary Of The Paper:**

The paper proposes a new method of updating deep neural networks for
combinatorial optimization problems during search using reinforcement learning.
In particular, the authors show that by updating only part of the network,
better results can be achieved at lower cost. They describe and evaluate their
method on different combinatorial optimization problems, comparing to other
machine-learning-based approaches as well as "traditional" solvers.

**Summary Of The Review:**

Interesting method with promising results.

---

> ### Author Response · Authors · 2021-11-22
> **Response**
>
> Thank you for reviewing our paper and for your very positive feedback! We are very happy that you believe that our method can have a large impact and that our evaluation is fair and convincing.
>
> Please let us address your remaining two minor points. Note that we already uploaded a new version of the paper that contains the mentioned changes.
>
> *Performance of Concorde*: You are wondering why the exact solver Concorde is so fast for the TSP. We are also often surprised by the performance of the (now 18 year old) Concorde code. The main reason for the good performance is that Concorde is highly-specialized on the TSP, and the instances we solve are, for the TSP, relatively small. It uses a cutting-plane method and many handcrafted components that cannot be applied similarly to other routing problems. In contrast, the heuristic solver LKH3 uses handcrafted heuristics that can be applied to a wide variety of non TSP-like routing problems like the CVRP. We made this more clear in the TSP results section.
>
> *Placement of new layer for EAS-Lay*: You are wishing for more insights on how to insert the new layer for EAS-Lay into the network. First, we want to emphasize that we noticed in our preliminary experiments that there are multiple layer placements that result in a good performance. In fact, we observed performance improvements over the original AS for almost all layer placements we evaluated. Nonetheless, some positions work better than others, and we extended the last paragraph of Section 3.2 to provide more details in one place. To summarize, the memory requirement of EAS-Lay can be reduced by inserting the additional layer closer towards the output layer of the network. This decreases  the number of layers to be considered during backpropagation. Furthermore, we noticed for transformer-based architectures that applying the residual layer to the query vector $q$ before it is passed to the single attention head usually results in a good performance.

---

> > ### Comment · Reviewer_JA4x · 2021-11-29
> > **Thank you**
> >
> > Thank you for your reply and clarifications!

---

### Public Comment · ~Buddhi_K1 · 2022-08-26
**Problem about tabular updates**

This is a good paper with a promising idea. I have some clarifications to make related to the concepts in section 3.3, tabular updates.

1. In section 3.3, you redefine the probability of selecting action $a_t$ in state $s_t$ as $q_\phi(a|s_t, \omega)^{\alpha} . Q_{g(s_t, a_t)}$. I could not understand the intuition behind this redefinition. Could you please clarify this?
2. Eq. (5) in section 3.3 is not also clear to me. I went through the (Dorigo et al. 2016) paper that you refer to, but I couldn't find a direct relation. Could you please clarify this too?

---

### Decision · Program_Chairs · 2022-01-20

**Decision:**

Accept (Poster)

**Comment:**

This paper gives a framework for using learning in combinatorial optimization problems.  In particular, active search is used to learn hueristics. The reviewers thought the paper had nice conceptual contributions for this approach and that the results would be very interesting to the community.